# Peer review of "Breast Cancer Care Quality Indicators in Spain: A Systematic Review"

_ijerph, 2021, doi:10.3390/ijerph18126411_

Round 1

Reviewer 1 Report

This is a very interesting systematic review that is needed for Spain. But there are some issues in need to be addressed prior to acceptance for publication:

  1. On line 74 the authors wrote "We rejected observational studies". Why??? This means that authors rejected prospective, retrospective and mixed cohort studies, rejected case-control and cross-sectional studies. This is very confusing and needs to be addressed in the text of the paper, as those articles should not be excluded from this review.
  2. On lines 88-90 the authors wrote "No suitable quality assessment instrument was available for this research topic. We developed a quality scoring system that captured all the QIs and specified the document." How did you validate the newly developed quality score? A detailed explanation will need to be included in here.
  3. On line 110 the authors wrote "Finally, only 15 were evaluated for full-text review". Is it 15 or 16? Your Fig. 1 shows 16.

Author Response

Response: First of all, we would like to express our sincere gratitude for all the comments and suggestions received from Reviewer 1. This information has undoubtedly enriched the text for its best understanding. We have clarified Reviewer1’s suggestions. We have introduced the required changes in our answers to the specific comments and the final manuscript. Changes are highlighted in bold type.

An English native speaker has thoroughly reviewed the text before submission of the revised manuscript. We hope that you will find the language of the manuscript acceptable. 

Reviewer 2 Report

This is an interesting paper on comparing the quality indicators for breast cancer management and treatment used in different areas of Spain with the EUSOMA document.

This is a challenging task as the heterogeneity in the operation of each breast unit is well known, hence the effort from EUSOMA to provide a standard set of QIs.

In order to make the scope of the paper clearer I would suggest to the authors to include a short description of the impementing framework of the EUSOMA document and to estimate the degree of adherence in each of the areas of Spain included in the paper, nased on their findings.

Also some background information on the organisation of breast cancer care in Spain would be helpful in the sense of describing the common elements among programmes and the degree of indepepndence in each local government to set their own priorities and to choose their own QIs. Is there any obligation to report some QIs for a national report on BC care? How is the performance of these units assessed at the national level?

How exactly do the authors propose to respond to the urgent need for establishing an agreed set of BC care QIs?

Some English language check is required e.g. line 43 beliefs NOT believes, line 372 QIs mostly collected NOT more collected, line 448 but nnot giving longer delay PLEASE EXPLAIN this is unclear.

Author Response

Thank you so much for appreciating our work. We are also thankful for all the recommendations suggested by Reviewer 2. We have made changes in our work taking them into account, and we are aware these modifications have improved our manuscript. As we did with Reviewer 1´s, the explanation of our revision is attached below, and additional material is in bold.

A revision of the entire manuscript has been undertaken for English language editing by an English native speaker.

Round 2

Reviewer 1 Report

The authors responded satisfactorily to all my comments.